**Data Availability Statement:** All relevant data are within the manuscript and its Supporting Information files.

**Funding:** This study was financially supported by European Commission [https://commission.europa.eu] in the form of a grant (PE00000013)

# Identifying technologies in circular economy paradigm through text mining on scientific literature

**Vito Giordano**[1,2]*, **Alessio Castagnoli**[1], **Isabella Pecorini**[1], **Filippo Chiarello**[1,2]

**1** Department of Energy, Systems, Territory, and Construction Engineering, Pisa, Italy, **2** Business Engineering for Data Science—B4DS Research Laboratory, Pisa, Italy

* vito.giordano@unipi.it

## Abstract

Technological innovation serves as the catalyst for the shift towards circular practices. Technologies not only address technical challenges, facilitating the transition to a more circular economy, but they also enhance business efficiency and profitability. Furthermore, they promote inclusivity and create job opportunities, ultimately yielding positive societal impacts. The research in this area tends to focus on digital technologies, neglecting other technological areas. Moreover, it heavily relies on literature reviews and expert opinions, potentially introducing biases. In this article we investigate the technological landscape of the circular economy through Natural Language Processing (NLP), examining key technologies used in this sector and the primary challenges in managing these technologies. The methodology is applied to more than 45,000 scientific publications and aims to extract technologies in the text of scientific articles with NLP. The findings of our analysis reveal a strong emphasis on emerging digital, life cycle assessment and biomaterials technologies. Furthermore, we identified seven distinct technological domains within the CE field. Finally, we provide advantages and problems arising in the adoption and implementation of these technologies in an industrial context.

## 1. Introduction

The Circular Economy (CE) paradigm has emerged as a pivotal strategy in the pursuit of sustainability and resource efficiency. As the global community tackles pressing environmental challenges, CE offers a promising path toward reducing waste, conserving resources, and fostering economic growth [1]. It is also noteworthy that CE endeavors to address social inequalities [2].

In this context, the technological aspect of CE assumes a paramount role. Technological innovation serves as the driving force behind the transition to circular practices. Over the recent years, there has been a growing body of research dedicated to investigate the CE and its practical application from a technological point of view. These studies mainly aim to reduce waste production through waste minimization technologies, such as 3D printing [3]. Digital

received by VG. This study was also financially supported by Erasmus+ (https://erasmus-plus.ec. europa.eu) in the form of awards (2023-1-IT02-KA220-HED-000158755 and ERASMUS-EDU-2021-PI-ALL-INNO (101055893)) received by FC. This study was also financially supported by Ministry of University and Research (MUR) [https://researchitaly.mur.gov.it] in the form of an award (PON 2014-2020 "Research and Innovation") received by AC.

**Competing interests:** The authors have declared that no competing interests exist.

applications have also been explored to enhance energy efficiency and traceability within the CE framework [4]. Other studies take a more comprehensive approach, attempting to understand overarching research themes, technological trends, and industrial initiatives in the CE domain. Many of these studies tend to focus on digital [5], information and communication (ICT) [6], or Industry 4.0 [7] technologies. [8] used emergent techniques, such as text mining, to review the scientific articles on CE and provide a map of current research topics.

Nevertheless, these researches have four main limitations. Firstly, it frequently lacks a comprehensive perspective on the CE, tending to concentrate on distinct technologies or specific industry sectors. Secondly, it emphasizes digital technologies, often overlooking other technological domains. Thirdly, the main research methodology relies heavily on literature reviews and expert opinions, potentially introducing biases. Lastly, even when automated techniques, such as text mining, are employed, the analysis tends to be narrow and may not thoroughly address the technological aspects of CE challenges.

Through advanced Natural Language Processing (NLP) techniques, a sub-field of text mining, we aim to overcome the above-mentioned limitations. We investigate three research questions in this study:

RQ1 –Which are the emerging technologies in CE?

RQ2 –Which are the main technological areas of CE?

RQ3 –How does this technological change impact at industrial level?

In this article, we examine the technological landscape of CE with NLP. The methodology is applied to more than 45,000 scientific publications and exploits NLP techniques to identify technologies in the text of scientific articles. This strategy enables us to track the rising interest in various technologies and, through network analysis, to uncover the connections between these technologies, thereby identifying the technological domains of CE.

The novelty of our paper lies in several key areas. First, unlike many existing papers that predominantly concentrate on individual technologies or industry domain in the CE topic, our paper presents a holistic view of the CE, addressing a broader spectrum of technological contexts.

Second, while a significant portion of the literature emphasizes the examination of digital technologies within the CE, leaving other technological domains relatively unexplored, our research expands beyond this narrow focus. By exploring a wide array of technological domains, we address the gaps left by existing literature and offer a more inclusive examination of CE technologies.

Third, our paper employs innovative research methods that move beyond traditional literature reviews, which often rely on expert opinions and can introduce biases. We utilize advanced automated techniques to enhance objectivity and depth in our analysis, thus mitigating potential biases.

Finally, even when scientific articles use automated techniques like NLP, their focus tends to be narrow and does not analyze the CE challenge comprehensively from a technological perspective. In contrast, our paper employs these automated techniques to provide a comprehensive technological analysis, addressing the broader implications and interactions within the CE landscape.

The findings of our analysis reveal a strong emphasis on emerging digital technologies. Additionally, we observe a growing research focus on technologies associated with lifecycle assessment and biomaterials. Furthermore, we have identified seven distinct technological domains within the CE field. Finally, we provide the key advantages and challenges pertaining to the adoption of these technologies within the CE framework and explain the main fields of potential application. In fact, RQ3 is designed to provide key practical implications by identifying the main issues and benefits associated with adopting these technologies. This focus

ensures that our findings are not only theoretically significant but also practically applicable, guiding industries in integrating these technologies to promote a circular economy effectively.

The identification of these technologies will significantly contribute to advancing the CE agenda in several ways. By comprehensively analyzing a broad array of technological domains within the CE, our research will uncover areas that have been underexplored or neglected. This identification of gaps in current research and technological implementation provides a clear roadmap for future investigations, ensuring that all technological aspects of CE are addressed and optimized.

Highlighting the technological advancements and their applications within the CE framework will inspire innovation across various sectors. Understanding how different technologies can be integrated and utilized in CE practices will encourage researchers and businesses to develop new solutions and improve existing processes. This innovation is crucial for achieving the efficiency and sustainability goals of the CE. Furthermore, the study can facilitate collaboration among different stakeholders. Researchers, businesses, and policymakers can use this information to work together on integrated solutions that span multiple sectors and technological domains.

A comprehensive understanding of the technological landscape in the CE will provide policymakers with the necessary insights to create informed and effective regulations and incentives. By identifying which technologies are most promising and which areas need more support, policymakers can tailor their strategies to promote sustainable practices, drive technological adoption, and ensure coherent and supportive regulatory frameworks. Moreover, the comprehensive analysis of CE technologies can support funding and investment decisions by providing evidence of the viability and impact of these technologies. Investors and funding agencies can use this information to allocate resources more effectively towards promising CE innovations.

The rest of the paper is structured as follows. Section 2 provides a description of the related works on CE and discusses the main NLP methods used to map a technological landscape. In Section 3, we explain the method we used. Section 4 shows the main insights and results of the paper. Section 5 proposes some implications that emerge from our work for researchers, practitioners and policy makers and details the main limitations of the proposed approach.

## 2. Related works

In this section, we firstly present background literature on CE in Section 2.1. In Section 2.2, we outline important papers on the identification of technologies from document text through NLP to provide the readers relevant information for fully understanding the method we adopted. Finally, we review papers that discuss topics similar to ours for delineating the main gaps in Section 2.3.

### 2.1 Circular economy: A complex paradigm

The evolution of the CE concept is based on two core principles [9]. The first is the waste hierarchy, defined by the European Commission [10] and now expanded as the 4R framework. The last defines the order of priority of operations aimed at proper waste management, i.e. Reduction, Reuse, Recycling and Recovery. The second key principle for the definition of a CE model is the system perspective, according to which a model can represent a macro-system (e.g. the whole system of human activity, a meso-system (e.g. eco-industrial parks) and a micro-system (e.g. an industrial process or part of it).

Developing a linear model towards a circular model means aiming to reduce waste generation and maximize resource recovery, primarily with a view to economic development and

improved environmental quality. Less explicitly, the CE aims to reduce social inequalities, often largely neglected as highlighted by [2].

A circular model does not necessarily equate to a sustainable development model. It represents a fundamental tool for its achievement, as waste generation and resource consumption often cause environmental, economic and social impacts [11]. [12] highlighted the relationships between sustainability and CE, among which it emerges how circularity is a necessary but not sufficient condition for the sustainability of a system and how it can be one of several solutions to achieve the latter.

This discrepancy between circularity and sustainability is reflected in the choices of societies. To date, they put profit before social and environmental impacts, encountering significant difficulties in solving the technological challenges of making processes efficient and sustainable at the same time. This confirms the strict need for regulatory intervention to properly support technological development [13].

A key role in the development of sustainable circular models will be played by digital technologies, aimed at automation and waste reduction along the supply chain [14], and by the bio-economy, which aims to make the best possible use of organic waste and replace some products with more easily recyclable bio-based alternatives [15]. In this context, the role of technology in CE is critically important. Technological innovations are at the forefront of shifting towards circular practices. The introduction and acceptance of cutting-edge technologies play a crucial part in making production methods more efficient, improving the use of resources, and supporting the principles of CE [16]. Additionally, advancements in technology that enable more effective recycling processes not only lessen the environmental burden but also generate employment opportunities in the recycling sector [17], which is in line with the aim of achieving social benefits. Similarly, the deployment of digital tools and intelligent technologies can make inclusivity and access better, ensuring the advantages of CE extend to a broader section of the population. Furthermore, the efficiencies brought about by technology, such as streamlined supply chains and minimized waste in manufacturing, result in cost reductions and higher profit margins for companies [12]. The creation and utilization of novel technologies related to CE also pave the way for new markets for unique products and services, promoting economic expansion and competitive strength internationally [1].

## 2.2 Mapping technological landscape with NLP

Various qualitative and quantitative methodologies exist for technological mapping. Qualitative approaches, like the Delphi technique and scenario mapping, often rely on expert judgment to identify upcoming trends. Conversely, quantitative methodologies usually employ mathematical models and statistical analyses, leveraging historical data to anticipate future trends.

The last mainly differentiates between two approaches: bibliometrics and NLP. Bibliometric methodologies use structured data such as the metadata of scientific publications (including authors, year of publication, and citations), or patent data (including assignee, inventor, citation count, and International Patent Classification). On the other hand, NLP leverages unstructured textual data to extract meaningful information. Despite its widespread use in the academic and professional community, bibliometric analysis has notable limitations. As demonstrated by [18], NLP outperforms bibliometric methods in capturing technological innovation phenomena.

One of the most applied NLP algorithm in technological mapping is topic modelling, and specifically Latent Dirichlet Allocation (LDA) which aims to extract topics and trace technological trends [19]. Recently, the focus of NLP literature has shifted towards word embedding

algorithms, which represent words as dense vectors of real numbers. Unlike LDA, which also aim to compute continuous representations of words, word embedding methods utilize artificial neural networks for word representation. [20] demonstrated that word embedding surpasses the LDA algorithm in preserving linear regularities among words and offers better computational efficiency than LDA when dealing with large text corpora. For instance, [21] employed word embeddings to develop a product landscape analysis to identify potential technology opportunities across multiple domains.

Recent advancements in NLP research have led to the development of a new class of techniques for representing words and phrases, known as transformers. Traditional word embeddings generate a single vector for a given word, regardless of its context. Transformers address this issue by factoring in the textual context of a word or phrase. One of the most well-known algorithms in this class is the Bidirectional Encoder Representations from Transformers (BERT) [22]. This model has been widely used in the innovation literature for various purposes, including the study of technological phenomena [23].

Current research on technological identification mainly targets the detection of emerging technologies rather than established ones, often leveraging patent classification (i.e., IPC code classes) [24] or applying greedy methods to textual data to extract general terms, not just technologies. [19], for instance, explored technology emergence using three text-based methodologies: tf-idf metrics for tracking technological shifts, LDA for assessing emerging topics, and a text-based score developed by [25]. These different methods offer unique perspectives, enriching the understanding of technological evolution. However, while their work yields significant insights, it encompasses not only technologies but also technology-related terms like "defective," "immune," and "delay fluorescence".

In a recent study, [26] overcame these limitations by designing a system capable of identifying technologies in patent documents using Named Entity Recognition, a branch of NLP. NER is a method that identifies and classifies entities into set categories text. Three NER methodologies applied in the literature are: (1) Gazetteer-based NER, which uses gazette, like Wikipedia, to map entities in text; (2) Rule-based NER, which constructs systems to extract specific entities using regular expressions and morphosyntactic information; (3) Distributional-based NER, which trains machine learning algorithms on manually annotated documents for entity recognition.

[26] used a combination of gazetteer, rule-based, and distributional-based methods to identify technologies in the text of patens. Their findings show that, in patent text, gazetteer methods yield high precision relative to other methods, while distributional methods possess higher recall. This system was applied by [27] to analyse a set of patents in the defense field. The goal was to map existing technologies and measure the level of technological convergence among different areas.

## 2.3 Trace the evolution of CE

In recent years, an increasing number of studies have focused on investigating the CE from a wide perspective. The topic was studied both from a purely technical perspective, concerning technologies and its implementations for the recovery of resources [28], and from an economic/managerial perspective to enable its integration into the current development model [29], as well as from a policy-making and regulatory perspective to enable the definition of a coherent framework characterized by limited environmental impacts and with the objective of a sustainable development model [30].

Focusing on the technological point of view, the literature is focused on studying current technological innovations to solve certain critical issues that exist today in the field of CE. In

fact, efforts have been made to reduce waste production through technologies like 3D printing [3] and the creation of bio-based materials [31]. These initiatives seek to both minimize waste at the source and enhance resource recovery throughout the production lifecycle. This involves refining existing processes via engineering solutions and digital tools, boosting energy efficiency, and integrating various operations through extensive data analysis [32]. The aim is also to tackle systemic issues such as product traceability [4]. Despite these advancements, a common critique of this body of work is its tendency to focus narrowly on specific technologies or sectors within the CE landscape. For example, [33] focused their analysis on the technological evolution of lithium batteries in a CE context, while [4, 34, 35] investigated the potential of blockchain technology in supporting CE principles.

In addition to the extensive research focusing on specific technologies, there are also studies that take a broader perspective, attempting to gain a comprehensive understanding of the key research topics and trends within this field. Numerous studies in this category have focused on the intersection between CE principles and digital or information and communication technologies (ICT). These investigations often employ literature review methodologies to elucidate various facets of this interdisciplinary field. For instance, [5, 36] systematically reviewed the existing literature to understand how digital technologies contribute to CE practices. [37] highlighted the key functions and mechanisms performed by digital technologies in advancing the CE agenda. Others concentrated on ICT as a solution to foster CE principles, emphasizing the interplay between the two. [6] delved into the significance of ICT in building a CE and examined successful practices within this framework. [38] extensively reviewed ICT solutions for the transition to a CE and categorized them based on technology and core CE concepts.

Some scholars explore the role of Industry 4.0 technologies in facilitating CE practices [39]. [40] conducted a survey of supply chain managers to investigate the adoption of Industry 4.0 technologies among companies and their intersection with CE principles. Similarly, [41] employed a survey on private manufacturing companies to provide a comprehensive perspective on the relationship between digital technologies, CE practices, and environmental policies during the COVID-19 pandemic period. However, even when researchers attempt to investigate a broader set of technologies often remain confined to digital, ICT, and Industry 4.0 technologies. This limitation significantly narrows the scope of the analysis, preventing a comprehensive examination of the full spectrum of technologies and their potential impacts on and interactions with the CE.

In addition to traditional literature review methods, some studies have sought to automate the analysis process using NLP techniques. NLP offers a way to address these challenges and provide more objective and efficient insights respect to traditional approach based on survey and case studies [42]. In CE field, [43] analysed 172 definitions of circular economy with the usage of this technique. [44] used NLP to comprehensively analyze two decades of research on waste management, while [45] employed NLP techniques, incorporating scientific papers and Twitter data, to gain a multifaceted understanding of waste management from different perspectives. Furthermore, [46] used LDA to conduct a systematic review of the literature pertaining to the CE in the context of cooling systems. [47] also applied NLP to review literature on non-technological climate mitigation solutions, proposing a taxonomy encompassing various solutions. Recently, [48] employed topic modelling based on word embeddings to review the literature in the field of resources, conservation and recycling.

However, similar to other cases, these studies tend to be concentrated on specific viewpoints or subfields within the CE. Only the research conducted by [8] combined NLP with the Delphi method to explore the academic viewpoint on the CE as a whole and did not delve into the analysis of a specific field. However, this work analyzes the field from a research perspective and not with a technological lens.

**Table 1. Overview of key studies that trace the evolution of CE.**

| Reference | Element of analysis | Level of analysis | Technology or Domain of Analysis | Research Method |
|---|---|---|---|---|
| [3] | Technological view point | Technology specific | 3D printing | Case study analysis |
| [33] | Technological view point | Technology specific | Lithium batteries | Literature review |
| [34] | Technological view point | Technology specific | Blockchain technology | Case study analysis |
| [4] | Technological view point | Technology specific | Blockchain technology | Narrative review |
| [35] | Technological view point | Technology specific | Blockchain technology | Literature review |
| [36] | Technological view point | Domain specific | Digital technologies | Literature review |
| [5] | Technological view point | Domain specific | Digital technologies | Literature review |
| [37] | Technological view point | Domain specific | Digital technologies | Literature review |
| [6] | Technological view point | Domain specific | ICT technologies | Literature review |
| [38] | Technological view point | Domain specific | ICT technologies | Literature review |
| [40] | Technological view point | Domain specific | Industry 4.0 technologies | Survey Analysis |
| [41] | Technological view point | Domain specific | Digital technologies | Survey Analysis |
| [43] | Broad view point | Broad Analysis | CE in general | NLP |
| [44] | Academic view point | Domain specific | Waste management | NLP |
| [45] | Academic view point | Domain specific | Waste management | NLP |
| [46] | Technological view point | Technology specific | Cooling systems | NLP |
| [47] | Non- Technological view point | Domain specific | Climate mitigation solutions | NLP |
| [48] | Academic view point | Domain specific | Resources, conservation and recycling | NLP |
| [8] | Academic view point | Broad Analysis | CE in general | NLP and Delphi |
| **Our Work** | **Technological view point** | **Broad Analysis** | **CE in general** | **NLP** |

Table 1 provides an overview of key studies in the field categorizing them based on their element of analysis, level of analysis, technology or domain focus, and research method. In the element of analysis, we shows the main viewpoint of each study, which can range from technological to academic or non-technological perspectives. The level of analysis distinguishes whether the study's analysis is technology-specific, domain-specific, or broad. Technology-specific analyses focus on particular technologies, while domain-specific analyses look at broader categories within the CE. Broad analyses provide a comprehensive view across multiple domains or technologies. We also show the technology or domain that each study investigates. Finally, we outlines the methodological approach taken in each study.

The table not only helps understand the scope and methods of various studies providing a summary of the background works but also highlights the differences between our work and others.

In summary, the main gaps in the existing literature, which aims to provide an overview of the CE landscape, can be outlined as follows:

1. Many papers predominantly concentrate on individual technologies or industry sectors, rather than presenting a holistic view of the CE;

2. A significant portion of the literature emphasizes the examination of digital technologies within the CE, leaving other technological domains relatively unexplored;

3. The prevalent research method employed is literature review, often relying on expert opinions. This approach may introduce potential issues and biases into the analysis;

4. Even when scientific articles employ more automated techniques like NLP, their focus tends to be narrow, and they do not analyze the CE challenge comprehensively from a technological perspective.

## 3. Methodological steps for identifying technologies

We used NLP systems to detect technologies within the titles, abstracts, and keywords of scientific publications on the CE. The process is composed of the following steps: (1) retrieving the collection of scientific papers; (2) identifying technologies through NER techniques; (3) cleaning and revising the data generated by the automatic identification process; and finally, (4) measuring the growth of technologies to understand emerging technologies in CE field and conducting network analysis to identify technological areas. The details of these steps are explained in this section.

### 3.1 Scientific papers retrieval

The objective of the initial phase was to acquire a collection of papers concerning the CE. These papers were sourced from Scopus. Our strategy for collecting papers was based on a keyword search. The selection of these keywords involved a two-step process. Initially, we examined significant papers and policy documents concerning the CE to identify keywords of importance as well as we rely on the definition of CE in Section 2. In Table 2, we present a list of the considered documents along with the keywords extracted from them. It is important to note that we paid particular attention to scientific papers as our search algorithm was primarily applied to these documents.

**Table 2. Documents used for identifying relevant keywords.**

| Type | Reference | Title | |
|---|---|---|---|
| Policy document | [49] | Global Resources Outlook 2019 | circular economy, waste prevention, resource recovery, secondary raw material, climate-neutral, by-product |
| | [50] | A sustainable bioeconomy for Europe | secondary raw material, resource recovery, bioeconomy, biorefinery, renewable resource, bio-based, ecological boundaries, added value product, biodegradable |
| | [51] | Circular Economy Action Plan | circular economy, waste reduction, waste prevention, waste recycling, product durability, remanufacturing, green transition, indicators |
| | Regulation (EU) 2020/852 of the European Parliament and of the Council of 18 June 2020 on the establishment of a framework to facilitate sustainable investment, and amending Regulation (EU) 2019/2088 | Establishment of a framework to facilitate sustainable investment, and amending Regulation | sustainability criteria, bioeconomy, circular economy, business strategy |
| | Directive 2010/75/EU of the European Parliament and of the Council of 24 November 2010 on industrial emissions (integrated pollution prevention and control) | Directive on industrial emissions (Integrated pollution prevention and control) | waste prevention, waste reduction, chemicals, waste labelling |
| Scientific paper | [52] | Towards circular economy implementation: A comprehensive review in context of manufacturing industry | Economic benefits, secondary raw material, waste recycling, waste prevention, ecodesign, resource scarcity, implementation |
| | [53] | Digitalisation as an Enabler of Circular Economy | Business model, digitalisation, ecodesign, industrial symbiosis |
| | [54] | Waste biorefinery models towards sustainable circular bioeconomy: critical review and future perspectives | bio-based products, biofuels, circular bioeconomy, biorefinery, life cycle assessment (LCA), renewable resource, low carbon technology |
| | [55] | Bioplastics for a circular economy | sustainability criteria, bioeconomy, renewable resource, ecodesign, waste prevention |
| | [56] | Industry 4.0 adoption and 10R advance manufacturing capabilities for sustainable development | sustainable development, advanced manufacturing, sustainability criteria, waste prevention, industrial symbiosis |

We also selected policy documents due to our interest in comprehending how technological shifts impact industry-level operations, a topic frequently discussed by policy makers. In the second step, three authors of this paper, each having expertise in the CE, reviewed the keywords collected from the first step. The goal of this review was to refine the list and retain only those keywords deemed relevant for searching the topic of interest within scientific publications.

Finally, the query used to retrieve the papers set of CE was:

TITLE-ABS-KEY("circular economy" OR "ecodesign" OR "eco design" OR "industrial symbiosis" OR "waste prevention" OR "waste reduction" OR "secondary raw material" OR "resource recovery" OR "sustainability criteria" OR "bioeconomy" OR "bio economy" OR "biorefinery" OR "bio refinery" OR "renewable resource" OR "waste recycling")

Our search was limited to the years between 2015 and 2023 (the query was launched in May 2023), as CE has gained considerable momentum and interest in recent years, and consequently, we ensured that our findings reflect the current state of the field. Moreover, we selected only journal papers and removed papers without abstracts since it is the main item of analysis. The query yielded 48,855 papers, inclusive of each document's title, abstract, keywords, and year of publication. The retrieved scientific articles can be accessed in the Supporting Information, in the S1 File.

## 3.2 Technologies identification with named entity recognition

Using NER, we were able to automatically detect technologies within the title, abstract and keywords of the 48,855 articles on CE. We established two distinct extraction methods based on gazetteer and rules-based principles to extract technological references from scientific papers. These methods were adapted from those proposed by [26, 27] for recognizing technologies within patent text.

For what concern gazetteer-based approach, two gazetteers were employed in this process:

- Wikipedia: this resource lists an array of emerging and potentially emerging technologies, totaling 397 different technologies;

- O*NET: an occupational framework created by the U.S. Department of Labor, ONET comprises 974 occupations categorized according to the Standard Occupational Classification (SOC) system, along with their corresponding skills, knowledge, abilities, and technologies. The framework encompasses 30,173 different technologies.

For the rule-based approach, we used a system known as Extractor4.0, as described in [26]. This system employs regular expressions to extract technologies 4.0.

We did not employ other machine learning methods and additional rule-based systems developed in [26, 27], because their effectiveness is heavily dependent on the nature of the text. In the original context, these methods were used on patent data. When applied to different types of text, such as academic, the performance of these methods may vary significantly.

Employing this methodology on the 48,855 scientific articles about CE, we extracted a total of 288,091 entities (including repetitions) and identified 5,182 unique entities. Specifically, 137 entities were recognized using the Wikipedia gazette, 906 were identified via the O*NET gazette, and 4,199 were extracted using the Extractor 4.0 system. Notably, only 58 of these entities were common across all NER systems.

## 3.3 Data pre-processing pipeline

The list of 5,182 entities was subsequently processed using a data pre-processing pipeline designed to remove non-technology terms and cluster synonymous and closely related

concepts. This process was guided by the approach outlined in [57], and it encompassed the following three steps: (1) Data cleaning; (2) Data Screening; and (3) Identifying synonyms.

During the (1) data cleaning process, we automatically eliminated:

- Entities composed of more than four words (e.g., "real-time not only the monitoring," "the world is accelerating the digitisation of many branches of the," "the complex way radio waves propagate," "gps aided geo augmented navigation," "v. with mass customized production becoming the," "modules and types of equipment").

- Entities composed of fewer than three characters (e.g., "nm," "mm," "v," "v.," "ie").

- Entities beginning with punctuation (e.g., ".net," ".networks," ".com," ".org," "-system").

Moreover, we conducted a manual review of the excluded words to prevent the omission of significant information. Following this step, all hyphen characters ("-") were replaced with an empty space to facilitate the identification of similar technologies (e.g., "cyber-physical-system" was replaced by "cyber-physical system"). After these modifications, the list of entities consisted of 2,742 items.

The (2) data screening was a manual process aimed at refining the corpus of entities extracted by the machine. The list of entities was analysed by two authors, experts in the CE field. Each author received a table containing the list of extracted entities, with the task being to determine whether each entity represented a technology for the CE field. The assignment was the following: "Read each extracted entity and decide whether the entity is a technology in the CE field or not". Authors relied on the definition of technology provided by [26] to decide whether an entity is a technology or not: "A technology is a technical mean or in general a technical system created by human-kind through the application of knowledge and science in order to solve a practical problem or perform a function" [26]. Moreover, they were permitted to consult external sources to assist in their evaluations. Following individual reviews, the authors convened to discuss any instances where consensus was not reached, and final decisions were made. Out of the total 2,742 entities, only 71 required collective deliberation, indicating a high level of agreement. After this process, the refined list comprised 1,542 technological entities.

The final step, (3) identifying synonyms, aimed to group synonymous technologies in our dataset using semantic similarity methods and pair-wise comparison. Each technology was translated into a vector in a 1024-dimensional semantic space using the Bidirectional Encoder Representations from Transformers (BERT) model [22]. The BERT model was employed to represent a technology as a 1024-dimension vector. Transforming technologies into vectors has been demonstrated as an effective method for identifying similarities between them, as shown recently by [58]. Next, we measured the cosine similarity between pairs of technology vectors, which is defined as the cosine of the angle between the two semantic vectors. Technologies with a cosine similarity exceeding 0.97 were grouped together, following the methodology proposed by [59]. We obtained a final set of 1,191 distinct technologies.

Our methodology leads to some technologies that are subsets of other technologies, and we chose to count both in our analysis. For example, 'computer' and 'notebook computer' are related, but they were treated as distinct technologies in our study. This approach was taken for several reasons. Firstly, the specificity of a technology might be important. 'Notebook computer' represents a more specific subset of the broader category 'computer,' and the differentiation could be relevant for certain analyses or applications. Secondly, treating them as separate entities allows for a more nuanced understanding of the technological landscape in the field of interest. By recognizing and retaining these distinctions, we can better understand the prevalence, importance, and context of specific technologies within the broader category.

### 3.4 Analysis of data

This section explains the main metrics and analyses that were conducted to address our research questions. Specifically, we employed two main strategies: (1) measuring the growth of technologies to discern the emerging technologies in the CE (to answer RQ1); and (2) conducting a network analysis to identify the main technological areas (RQ2).

To identify technological trends in CE scientific publications, we assessed the growth of each technology. Our main assumption is that a swift increase in the number of papers indicates a technology with high growth potential [60]. A considerable body of literature outlines methods for measuring technology growth, with growth curves (or S-curves) being the most commonly utilized [61]. However, these methods don't provide a synthetic growth index for a technology (which is what our present method requires), but rather indicate the technology's diffusion stage (emerging, growth, maturity, or saturation phase). Technology trend analysis (TTA) is another methodology employed to comprehend a technology's growth pattern in the literature. [62] review the literature on approaches and methodologies used for TTA. Among these, we employed a synthetic index: the Relative Development of Growth Rates (RDGR). The RDGR was first introduced in the seminal paper by [63], and it is extensively used in literature to evaluate technology growth. For instance, [64] in assessing the technological attractiveness of cloud computing employed the RDGR. [57] also used the RDGR to measure if and how much the European Skills, Competences, Qualifications and Occupations (ESCO) is updated with technological trends in scientific publications.

For each technology, we computed the RDGR to examine the growth rate of technologies as reflected in scientific papers. We determined the RDGR in accordance with the method suggested by [57]. The RDGR is calculated by taking the number of papers published during the most recent three-year period (i.e., 2021–2023) and dividing it by the number of papers published over a preceding six-year span (i.e., 2015–2020):

$$RDGR_i = \frac{(N.\ of\ papers\ containing\ Tech_i)_{2021-2023}}{(N.\ of\ papers\ containing\ Tech_i)_{2015-2020}}$$

The formula considers the portion of scientific literature about the $i$-th technology produced in the last 3 years on the total number of articles. Calculating the RDGR index may not be feasible for some technologies due to a lack of papers in specific years. If the number of papers in the last three years is zero, the numerator of the RDGR index becomes zero. Similarly, if no papers were published in the first six years, the denominator becomes zero. Technologies with no papers in the first six years (2015–2020) but with publications in the last three years (2021–2023) were classified as "emergent technologies". Conversely, we defined "obsolescent technologies" as those with papers published in the first six years but none in the later years are.

To identify the main technological areas in CE, we performed a network analysis. This grouped CE-related technologies into clusters based on their co-occurrence in our paper set. In this analysis, each technology is a node, with edges established based on co-occurrences in the same scientific article. The edge thickness and node proximity depend on the co-occurrence, while the node size corresponds to its degree—the sum of edge weights connected to it. This measure, known as degree centrality, quantifies a technology's centrality in the network, helping to identify technologies which are central pillars within the CE field and their clusters. We excluded clusters of technologies of two nodes without connections to the main graph, prioritizing the most interconnected technologies within the CE field. As a result, our network analysis considered a total of 949 technologies (out of 1,191).

The network analysis was conducted using the Louvain algorithm [65]. This resulted in a modularity of 0.410 and identified 7 clusters. Modularity is a measure that quantifies the

strength of division of a network into clusters, ranging from -0.50 and +1.00. A high modularity score indicates that the network's nodes are organized into tightly-knit groups, with dense connections within each group and sparser connections between groups. The Louvain method is particularly suitable as it autonomously identifies the number of communities into which the network can be partitioned.

The complete list of technologies with RDGR,the clusters of technologies, as well as the set of scientific papers connected to each technological cluster are available in the supporting informaion. Specifically, the retrieved articles can be accessed in the S1 File. The list of all technologies with the growth index (RDGR) can be accessed in the S2 File. The clusters of technologies with all information connected can be accessed in the S3 File.

## 4. Results and discussion

### 4.1 Fast growing technologies in CE

In this section, we present the results of our analysis of emerging technologies in the field of CE. Our research question (RQ1) aimed to identify which technologies are considered to be emerging within this field. To answer this question, we analysed the most growing technologies in terms of RDGR.

Table 3 presents detailed information about the top-15 fastest-growing technologies identified using our NLP method. For each technology, the table includes several key metrics. The total number of papers indicates how many scientific papers have mentioned or discussed the technology in the context of the CE. The percentage of papers shows the proportion of papers on this specific technology relative to the total number of papers we retrieved. Additionally, the table provides the number of papers published in two distinct periods: from 2015 to 2020 and from 2021 to 2023. This division allows us to see the historical and recent interest in each technology. The RDGR measures the growth rate of the technology. It is calculated by dividing the number of papers published in the last three years (2021–2023) by the number of papers published in the preceding six years (2015–2020), as explained in Section 3.4. A higher RDGR indicates a faster-growing technology.

**Table 3. Fast growing technologies.**

| Technology | No. of Papers | % of Papers | No. of Papers (2015–2020) | No. of Papers (2021–2023) | RDGR |
|---|---|---|---|---|---|
| digital twins | 19 | 0.04 | 1 | 18 | **18.00** |
| deep learning | 47 | 0.10 | 8 | 39 | **4.88** |
| blockchain | 139 | 0.28 | 24 | 115 | **4.79** |
| machine learning | 181 | 0.37 | 38 | 143 | **3.76** |
| personal protective equipment | 30 | 0.06 | 7 | 23 | **3.29** |
| artificial intelligence | 152 | 0.31 | 39 | 113 | **2.90** |
| fused filament fabrication | 16 | 0.03 | 5 | 11 | **2.20** |
| life cycle impact assessment | 37 | 0.08 | 12 | 25 | **2.08** |
| life cycle sustainability assessment | 36 | 0.07 | 12 | 24 | **2.00** |
| robot | 47 | 0.10 | 16 | 31 | **1.94** |
| internet of things | 169 | 0.34 | 59 | 110 | **1.86** |
| additive manufacturing | 124 | 0.25 | 44 | 80 | **1.82** |
| quantum dots | 36 | 0.07 | 13 | 23 | **1.77** |
| cyber physical system | 22 | 0.04 | 8 | 14 | **1.75** |
| bioplastic | 113 | 0.23 | 42 | 71 | **1.69** |

To better understand how to read Table 3, consider the example of the technology "Digital Twins." This technology is mentioned in 19 scientific papers, which constitutes 0.04% of all papers focused on CE. Between 2015 and 2020, only one paper discussed digital twins. However, between 2021 and 2023, this number increased to 18 papers. The RDGR for digital twins is 18.00. This means that in recent years (2021–2023), the number of papers discussing digital twins is 18 times higher compared to the previous years (2015–2020). The example demonstrates that initially, digital twins were scarcely mentioned in the literature (only one paper from 2015 to 2020). However, its prominence has sharply increased recently, with 18 papers published from 2021 to 2023. The high RDGR underscores its rapid emergence and growing importance.

This table provides a comprehensive overview of the growth trends of these technologies over time. By including these specific metrics, it allows for easy comparison between different technologies, highlighting which ones are emerging more rapidly in the field of the circular economy.

Between the Fast growing technologies could be identified three different groups of technologies: "Digital technologies", "New process and materials" and "Assessment technologies".

The first one includes technologies with a higher growth factor, characterized by a high pervasiveness. The principal technologies are *Machine learning* and *Internet of things*. They are largely employed in data analysis and process efficiency, as they are used to control processes and parameters in every field [66]. This group includes the most growing technology *Digital twins*, which offer a powerful tool for optimizing resource use, enhancing transparency, and supporting decision-making in the CE. They enable a more holistic and data-driven approach to design, production, and consumption, fostering a transition towards a more sustainable and circular system.

The second one includes new materials and new industrial processes. Materials are exploited for the creation of electrodes for recovery/purification processes, as materials obtainable from waste sources or as materials destined for the recycling chain. The group is characterized by a strong presence in the bio-economy. Particularly extensive is the study of bioplastics, studied both in terms of their impact as a substitute material for plastics and as a material present in the organic residual fraction, trying to define their role in recovery processes [67]. The second most important technology is *additive manufacturing*, which helps to minimize resource consumption, reduce waste, extend product lifespans, and promote a more sustainable and efficient approach to manufacturing and consumption. Particular importance is given to the role of the resources employed, with a strong study on the use of *bioplastics* such as polylactic acid (PLA) and polyhydroxybutyrate (PHB).

The last group includes *Life Cycle Sustainability Assessment* and *Life Cycle Impact Assessment*. The first is an assessment technology that integrates the three pillars of sustainability into a single study, linking economic, environmental and social sustainability. It is exploited in numerous fields, such as renewable energy, bio-economy, ecodesign and hydrogen [68]. The second one is a methodology used to evaluate the potential environmental impacts of a product, process, or activity throughout its entire life cycle. It is a crucial step in the life cycle assessment (LCA).

## 4.2 Technological areas in CE

In this section, we present the results of our analysis of the main technological areas within the field of CE. Our research question (RQ2) aimed to identify the most relevant and prominent technological areas within this field. To answer this question, we conducted a network analysis of technologies related to the CE, and grouped them into different clusters based on their occurrences in the paper set.

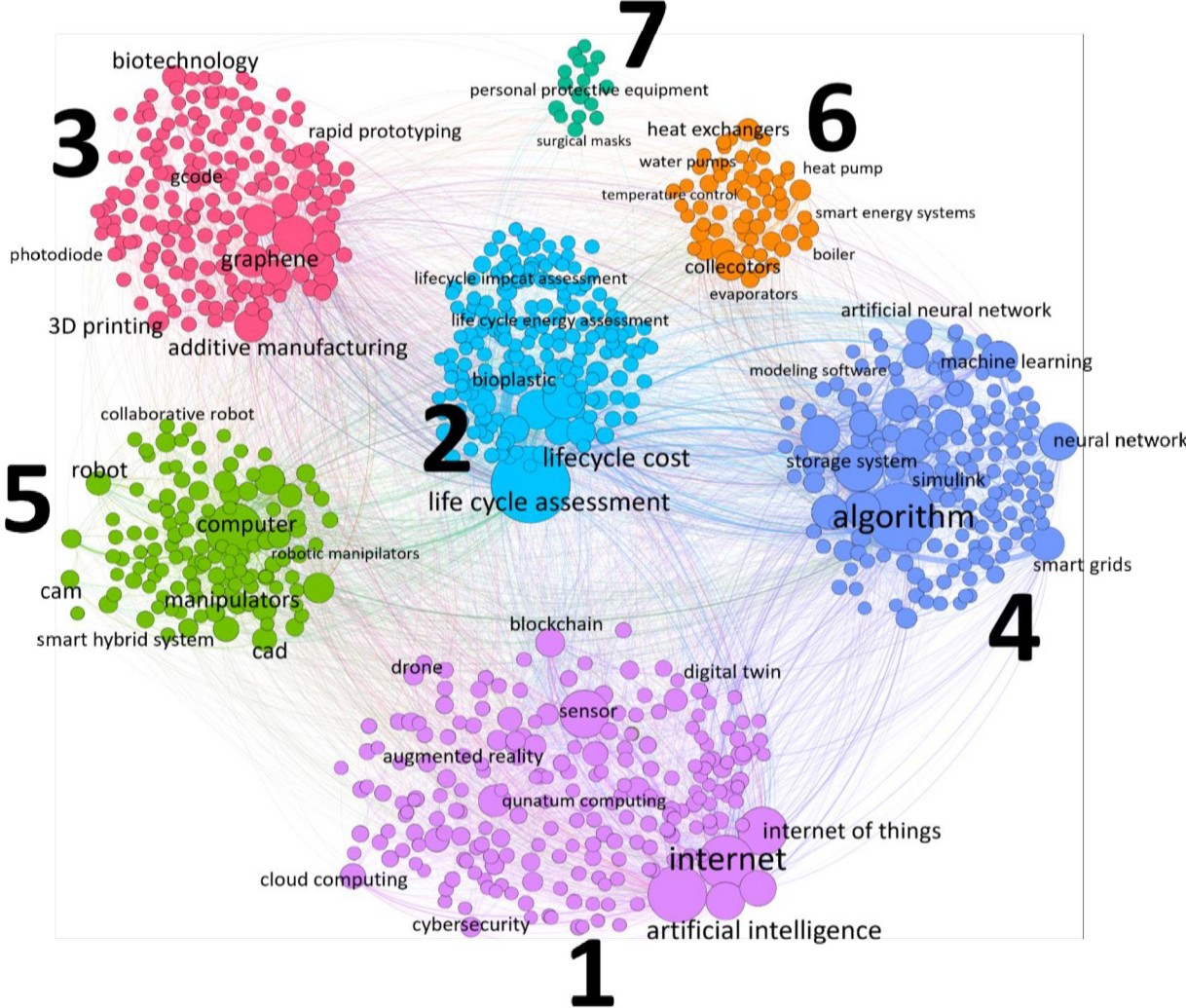

**Fig 1. Clustering of technologies in CE.**

Fig 1 provides a visual representation of the clustering of technologies related to the CE. Each node in the graph represents a specific technology identified in our analysis. The size of a node indicates the centrality or importance of the technology within the network. Centrality in this context is calculated using a specific metric known as degree centrality. The degree centrality measures the number of direct connections a node (technology) has to other nodes in the network. In other words, it counts how many times a specific technology co-occurs with other technologies in the scientific papers analyzed. Larger nodes represent technologies that are more central and interconnected with other technologies. The lines connecting the nodes are known as edges. They represent the co-occurrence of technologies within the same scientific papers. The thickness of an edge indicates the strength of the connection between two technologies. Thicker edges mean a higher frequency of co-occurrence. The nodes are grouped into clusters, which are color-coded. Each cluster represents a distinct technological area within the CE field. The clustering algorithm used (Louvain method as described in Section 3.4) identifies these groups based on the strength of connections within the network, ensuring that technologies frequently mentioned together are grouped into the same cluster.

**Table 4. Clustering of technologies in CE.**

| # | Cluster | Size | No. of Obsolescent Techs | No. of Emergent Techs | Mean RDGR | SD | Top-15 Technologies (Degree centrality) |
|---|---|---|---|---|---|---|---|
| 1 | Applied computer science | 217 | 53 (24%) | 60 (28%) | 1.43 | 2.18 | sensors (189); internet (177); internet of things (150); sensor (149); artificial intelligence (106); platforms (97); wireless (81); blockchain (66); cloud computing (52); energy harvesting (50); cameras (47); robotics (46); actuators (44); data mining (42); remote sensing (40) |
| 2 | Assessment Technologies | 182 | 57 (31%) | 46 (25%) | 0.97 | 0.57 | life cycle assessment (273); prototype (123); database (115); life cycle costing (72); decision support system (65); statistics (65); retrofits (63); product life cycle (62); python program (33); processor (26); user interface (25); automobiles (21); combustion engines (21); plc (19); bioplastic (18) |
| 3 | Advanced materials and manufacturing technologies | 179 | 49 (27%) | 34 (19%) | 1.11 | 0.76 | catalyst (115); laser (86); additive manufacturing (84); reactors (79); semiconductor (55); biotechnological (50); graphene (50); filters (49); supercapacitors (48); electrodes (48); biotechnology (45); supercapacitor (44); prototyping (39); bioreactors (38); 3d printers (33) |
| 4 | Intelligent systems and energy optimization | 165 | 49 (30%) | 34 (21%) | 0.84 | 0.82 | algorithm (235); simulations (148); storage system (107); neural networks (105); machine learning (100); controller (94); grids (94); smart grids (92); generators (92); control systems (90); controllers (60); bus (59); electric vehicle (57); aspen plus (53); artificial neural network (52) |
| 5 | Hardware | 128 | 36 (28%) | 29 (23%) | 1.07 | 0.68 | computer (179); electronic devices (73); smartphones (69); mobile phone (52); robot (48); cad (47); laptop (44); computers (44); televisions (38); washing machines (37); robots (34); lcd (33); monitors (31); intelligent devices (29); processors (28) |
| 6 | Thermal energy technologies | 59 | 14 (24%) | 9 (15%) | 0.77 | 0.58 | collectors (70); boilers (42); heat exchangers (41); heat pumps (39); district heating (32); trucks (25); smart energy systems (24); absorbers (21); evaporators (20); furnaces (20); water heaters (20); fans (15); storage tanks (14); condensers (13); ventilation systems (10) |
| 7 | Single-use products | 16 | 1 (6%) | 11 (69%) | 3.57 | 3.1 | personal protective equipment (20); face masks (13); surgical masks (9); protective clothing (7); respirators (6); fluorescent lamps (4); parachutes (4); welders (4); bullet proof vests (4); safety goggles (3); goggles (3); visors (2); manufactured without chemical additives (1); syringes (1); face shields (1) |

Fig 1 and related Table 4 show the results of network analysis performed on technologies extracted from scientific papers. Each cluster is shown in Fig 1 using different colors.

The content of each cluster is shown in Table 4, where a comprehensive overview of the technologies within each cluster is presented. Each cluster is given a label based on the most common and relevant terms within the cluster. These labels are derived from the prominent tokens or terms occurring in each cluster but are manually assigned by the authors to ensure they accurately reflect the thematic focus of the technologies grouped together. This careful labelling helps readers quickly understand the key technological themes represented in each cluster. For each cluster we can see the label of the cluster, the number of technologies within the cluster (Size), the number of technologies that are considered obsolescent (No. of Obsolescent Techs), the number of technologies that are considered emergent (No. of Emergent Techs), the mean RDGR of the technologies (Mean GI), the standard deviation of the RDGR (SD), the top 15 technologies based on degree centrality measure, so technologies that have the highest number of connections to other technologies in the network, making them the most central and influential within their cluster.

To determine if a technology is obsolescent, we look at its RDGR. Technologies with an RDGR lower than 1 and close to 0 are considered obsolescent. This low RDGR indicates that the number of papers discussing these technologies has decreased significantly in recent years compared to previous years, suggesting a decline in their relevance or usage. Conversely, if the

RDGR is higher than 1 and tends to be very high, the technology is considered emerging. A high RDGR indicates a significant increase in the number of papers discussing the technology in recent years compared to previous years, signifying a growing interest and relevance in the field. For example, technologies like "Digital Twins" with an RDGR of 18.00 demonstrate rapid growth, highlighting their emerging importance within the circular economy.

Similarly, the mean of RDGR related to each technological cluster provides an average growth rate for the technologies, indicating how rapidly the technologies are gaining attention. This measure gives us an overview of whether the technological cluster is obsolescent or emerging and how quickly the cluster is growing on average.

By analyzing Fig 1 and Table 4 together, you can gain a detailed understanding of how various technologies are interconnected and how they are grouped into thematic clusters. This combined analysis highlights the central technologies driving the CE and identifies both emerging and declining trends within the field.

The results of the network analysis are presented in a visual format and can be accessed through the following link: https://vitogiordano94.github.io/Technologies-in-Circular-Economy/network_analysis/. This link will take you to a webpage where you can view the network and interact with it. In the link, the reader can zoom in and out using his/her mouse scroll wheel or the webpage's zoom controls and pan across the network by clicking and dragging the background. The reader can also search a specific node (technology) in the searching box to see detailed information about the technology it represents, and click on a node to fix its position and highlight its connections.

The first one is "Applied computer science", characterized by technologies related to data capture and transmission. *Sensors*, *Internet* and *Internet of things* are the most important technologies, with centrality degrees of 189, 177 and 150 respectively. The 60 emerging technologies, representing 28% of the size of the whole cluster, and the mean RDGR of 1.43 indicate the high growth of this cluster. This cluster is also characterized by a high number of obsolescent technologies, showing the rapid evolution that characterizes digital technologies. The *Internet of things* is also an emerging technology, highlighting the growing importance of web applications to real time monitoring of data and the strong correlation between digitalization and circularity [69]. *Sensors* find application in multiple fields, from real-time monitoring of processes to in site monitoring of physical-chemical parameters, images, chemistry and others [70]. A considerable number of papers focus on the study of new sensors for use in various fields of the CE, such as water treatment or the monitoring of reactor parameters. Strong emphasis is also placed on the widespread use of sensors for the development of smart processes, which are used for production efficiency. In this topic, we find even the *internet* and *internet of things*, widely used for communication and sharing of data, necessary for processes and practice optimization, such as smart irrigation [71].

The second one is "Assessment technologies", characterized by decision-support tools, statistics application and prototyping. The most important technologies in this cluster are *Life cycle assessment* (LCA), *Prototype* and *Database* with respective centrality degrees of 273, 123 and 115. It includes 46 emergent technologies and 57 obsolescent technologies, with a mean RDGR of 0.97. LCA is characterized by its high pervasiveness, as it finds application in numerous fields as an assessment. It is used as an evaluation tool to identify environmental inefficiencies of a system, as a forecasting tool to evaluate the different impacts of two or more approaches and as a support tool for administrations for the elaboration of plans and programs [72, 73]. It is applied in the sector of waste prevention and recovery, in the field of renewable energies and in the field of remediation [74]. It is often integrated with other evaluation technologies, such as *Life life cycle Costing* or *Social LCA*, making it possible to identify any trade-offs deriving from specific scenarios [1, 75]. For proper evaluation and creation of correct decision

support tools, *databases* are of paramount importance. They are essential tools for data management, tracking resources, promoting transparency, supporting decision-making, and fostering collaboration in the implementation of CE principles and practices [76]. They enable the efficient management of resources, encourage sustainable production and consumption, and contribute to the overall sustainability and resilience of economic systems [77]. Finally, under the heading prototype, a vast range of different applications can be identified, due to the rather general meaning of the term. They are used as a pilot plant of real reactors, as in the case of biorefineries, as prototypes of components or modules for solar panels or as prototypes of sustainability assessment tools.

The third cluster is denominated "Advanced materials and manufacturing technologies". The three most important technologies are *Catalyst*, *Laser* and *Additive manufacturing*, with centrality degrees of 115, 86 and 84. The cluster's technologies are strongly applied in the field of bio-economy and chemical process efficiency. *Catalysts* are mentioned both in research aimed at the development of new catalysts from renewable raw materials and in their use for the optimization of bioprocesses such as anaerobic digestion to obtain new bioproducts from organic waste [78, 79]. These processes are appropriately implemented within reactors, which are particularly designed to optimize existing processes through the implementation of new technologies or the development of new processes for the recovery of organic waste [80]. In this sense, all technologies related to integrated biorefinery are applied, with the study of processes based on both biochemical and thermochemical conversion, such as gasification [81]. *Lasers* have a variety of applications, both for the production of specific materials used as catalysts and as part of analytical instrumentation such as mass spectrometry [82; 83].

"Intelligent Systems and energy optimization" is the fourth cluster per size. Its most important technologies are *Algorithm*, *Simulation* and *Storage system*, with centrality degrees of 235, 148 and 107 respectively. *Algorithms* are used to optimize various aspects of energy systems, such as load scheduling, resource allocation, and energy management [84]. *Simulations* are used to model and simulate the behavior of energy systems, including power grids, smart grids, and renewable energy systems. These simulations provide insights into system dynamics, allow testing of different scenarios, and help optimize system configurations and operations. They can determine the optimal charging and discharging schedules, taking into account factors such as electricity prices, grid conditions, and energy demand patterns [85]. Through their application, intelligent systems can make data-driven decisions, optimize energy usage, improve grid stability, and enhance overall energy management and efficiency. These tools enable better planning, operation, and control of energy systems, ultimately supporting the transition to a more sustainable and optimized energy future [86] *Storage systems* are the technologies involved in the electrification process in countries, the first one is aimed at a more efficient distribution of energy in countries through the implementation of sensors and digital technologies that allows the optimization of electricity transports and the reduction of energy loss [87]. *Storage systems* are instead aimed at the storage of produced energy, by energy plants or by energy communities [88, 89].

"Hardware" is the fifth cluster for size. It includes the most common instruments for everyday work, highlighted by the three most important technologies: *computer*, *electronic devices* and *smartphones* with centrality degrees of 179, 73 and 69. Even this cluster is characterized by a higher number of obsolescent technologies than emerging technologies, pointing to the lack of development in research activities about the technologies involved. *Computers* find very large applications, due to their implication in every technology that involves data acquisition, storage, analysis and modelling [90]. They find use in every field of CE, for example for models aimed at the optimization of biorefineries and the prediction of the environmental impacts related to the component design by the computer-aided tool [91]. Moreover, together with

*smartphones*, they represent two of the main sources of e-waste. This is why they are mentioned as objects of recycling processes for the recovery of components and the study of more sustainable configurations from an ecodesign perspective [92].

"Thermal energy" is the sixth cluster. The most important technologies are *Collectors*, *Boiler* and *Heat exchangers*, respectively with centrality degrees of 70, 42 and 41. Is characterized by the less mean RDGR and the less difference between emergent technologies and obsolescent technologies (-9%), indexing the stability reached on this topic. *Collectors* find applications in both energy recovery from waste and energy efficiency in buildings and plants. Both through an assessment of their impact on processes and from a development perspective, through the use of new technologies and materials. Energy performance evaluation and improvement studies are also found for heat exchangers [93, 94]. Furthermore, papers dealing with the mapping of the territory where heat pumps can be applied are widespread [95].

The last cluster per size is "Single-use products", characterized by the highest RDGR but also a few papers. Technologies are relegated to the healthcare and laboratory fields, for which there has been increased interest in recycling and waste prevention following the COVID-19 pandemic [96]. The first three technologies are personal protective equipment, face masks and surgical masks, with a centrality degree of 20, 13 and 9 respectively. Considerable emphasis is placed on the potential applications of bioplastics for the production of these devices, which would facilitate their recycling and reduce pollution from microplastics and nanoplastics [97].

## 4.3 The impact of technological change at industrial level

Answering RQ3 helps us understand the practical implications of the identified technologies, focusing on the main issues and benefits of their adoption. This section delves into how these technologies can be practically implemented in industries or integrated into existing systems to promote the circular economy. By examining the real-world impact of these technological changes, we aim to provide clear guidance for industries looking to adopt these innovations.

Besides being the largest, "Applied computer science" is the most pervasive cluster. Its technologies find application in all fields of the CE, in particular in waste management and process efficiency through the application of the industry 4.0 concept [69, 98]. The advantages of their application are numerous, which can be summarized mainly in the efficiency enhancement of processes, the possibility of knowing and exploiting data in real-time and obtaining greater transparency and reliability in the supply chain, thanks also to support through data-driven processes [99]. Their application, however, entails some problems, such as the energy consumption for data exchange and accumulation [100] and the lack of shared standards and frameworks that make compliance with regulations more complex for some sectors [101].

The second cluster in terms of size is also very pervasive, as *Life Cycle Thinking* ubiquitously applied in the fields of the CE. The advantage of these applications is the possibility to support decisions objectively and to clearly and comprehensively explain a large amount of data [102]. Decision support takes place both during the development of products and processes and for the analysis of planning or technological scenarios, finding a fundamental application in the development of new technologies for waste recovery [103]. The disadvantage concerning only environmental or economic aspects evaluated, respectively for LCA and LCC, is as described above being overcome as the application of LCSA is increasingly emerging [16]. As for the variability of the data used and the not-always-transparent results, the European Commission is working on the implementation of PEF and OEF, certifications for which it is compulsory to use a specific dataset and follow a stricter procedure than the ISO technical standard of reference [104]. *Bioplastics* offer several advantages compared to traditional petroleum-based plastics. They derive from renewable resources, such as plants, agricultural by-products, or algae,

reducing our dependence on fossil fuels and decreasing carbon emissions [105]. Additionally, some *bioplastics* are designed to be biodegradable or compostable, enabling them to break down naturally and reduce waste and pollution [106]. *Bioplastics* also contribute to resource efficiency by promoting the use of renewable feedstocks and reducing the extraction of finite resources. They can be part of a closed-loop system, where bioplastics are collected, recycled, and reused, contributing to the circularity of materials [107]. Moreover, *bioplastics* can help address the issue of plastic pollution by providing eco-friendly alternatives that have a lower environmental impact. Overall, *bioplastics* offer a more sustainable and environmentally friendly option in the CE, supporting the shift towards a more sustainable and resource-efficient future [108].

The cluster "Advanced materials and manufacturing technologies" improves the circularity of numerous fields of application. *Fused Filament Fabrication* and *additive manufacturing* technologies provide design freedom, waste reduction, and customization capabilities [109]. Quantum dots offer unique optical properties with potential applications in multiple industries but require addressing concerns related to toxicity and scalability [110]. *Fermentation systems* enable waste valorization, resource efficiency, and diverse applications but face challenges related to feedstock availability, process optimization, downstream processing, and market acceptance [111, 112]. Despite their advantages and criticalities, these technologies contribute to the CE by promoting sustainable practices, reducing waste, and enabling efficient resource utilization in various fields [113]. Continued research and development efforts are essential to overcome the criticalities and maximize the potential of these technologies in the CE.

"Intelligent systems", which encompass a range of technologies including *deep learning*, *automation*, and *machine learning*, are reshaping industrial operations [114]. These systems can perform complex tasks with precision, speed, and consistency, leading to increased productivity and reduced costs [115]. *Intelligent robots* and *automation systems* streamline manufacturing processes, optimize supply chain management, and enhance quality control. *Energy optimization* technologies are becoming essential in industrial settings to enhance energy efficiency, reduce carbon emissions, and optimize resource utilization [116]. Through real-time monitoring, data analysis, and intelligent control algorithms, energy optimization systems can identify areas of energy wastage, recommend energy-saving measures, and optimize energy usage based on demand and supply. This results in significant energy savings, cost reductions, and environmental benefits [117]. Industries can adopt *energy optimization technologies* to achieve sustainability goals, comply with regulations, and enhance their competitive advantage in a resource-constrained world. Collectively, these emergent technologies have the potential to revolutionize industrial operations, leading to increased productivity, improved efficiency, reduced costs, and enhanced sustainability [118]. By embracing *deep learning*, *machine learning* and *supercapacitors*, industries can stay ahead of the technological curve, adapt to changing market dynamics, and unlock new opportunities for growth [30]. However, it is important to address challenges related to implementation, such as infrastructure requirements, data privacy concerns, and skill gaps. With proper planning, investment, and collaboration, these technologies can drive the industrial sector towards a more efficient, sustainable, and intelligent future [69].

Emerging technologies related to the "Hardware" cluster are playing pivotal roles in driving technological advancements and enabling CE principles in industrial sectors. *Intelligent devices* facilitate real-time monitoring, data collection, and process automation, leading to enhanced productivity, efficiency, and resource optimization [119, 120]. By connecting machines, equipment, and systems, intelligent devices enable predictive maintenance, reducing downtime and improving asset management. They also contribute to energy optimization by providing insights into energy consumption patterns, enabling companies to identify energy-saving opportunities and optimize their operations accordingly. In industrial settings, *computer vision*

systems enhance quality control, inspection processes, and object recognition tasks. They can identify defects, anomalies, or irregularities in products or materials with high precision and speed, reducing the need for manual inspection. This improves product quality, reduces waste, and enhances overall process efficiency. Moreover, *computer vision* systems enable automated sorting and recycling processes, contributing to CE objectives by enhancing waste management and resource recovery [121]. Overall, these emerging technologies are driving significant technological and circular changes in the industrial sector. These technologies enhance efficiency, productivity, and resource optimization, while also enabling sustainable practices such as waste reduction, energy optimization, and material reuse [122].

The use of "single-use products" leads to serious resource consumption and environmental impact problems, mainly due to their plastic nature [123]. Therefore, a strong role will be played by the development of systems that enable the reduction of their impact on the environment and the creation of closed-loop cycles.

## 5. Conclusions

In this study, we have utilized advanced NLP techniques to investigate the technological landscape within the CE paradigm. Our analysis of over 45,000 scientific publications has provided significant insights into the emerging and prevailing technologies driving the transition towards a more sustainable and circular economy.

The findings revealed a strong emphasis on digital technologies, life cycle assessment methodologies, and biomaterials. Through the identification of seven distinct technological domains, we were able to categorize the various technological advancements and their applications within the CE field. These domains include Digital Technologies, New Processes and Materials, and Assessment Technologies, each playing a crucial role in enhancing resource efficiency, reducing waste, and promoting sustainable industrial practices. Our network analysis, visually represented in Fig 1 and detailed in Table 4, illustrates the interconnectedness of these technologies and their relative importance within the CE landscape. Technologies such as Digital Twins, Machine Learning, and Internet of Things (IoT) were identified as highly central and rapidly emerging, signifying their increasing relevance and application across various sectors. On the other hand, technologies with lower RDGR were considered obsolescent, indicating a decline in their significance.

One of the key contributions of our research is the development of RQ3, which specifically addresses the industrial implications of these technological changes. By examining how these technologies can be practically implemented and integrated into existing systems, we have provided valuable insights into their potential to promote the circular economy. In Section 4.3, we highlight specific insights into the advantages and challenges of adopting these technologies at an industrial level. For instance, Digital Twins, identified as an emerging technology with a high RDGR, offer significant benefits such as improved efficiency, enhanced decision-making, cost savings, and sustainability. However, they also pose challenges including data accuracy and integration, data security, and organizational changes. Similarly, Virtual Reality technology provides immersive experiences and resource-saving opportunities but faces issues related to motion sickness, cost, and accessibility. These detailed insights into each technology's advantages, challenges, and fields of application underscore the practical implications and guide industries in effectively integrating these technologies to advance the circular economy.

The paper offers several valuable contributions from the viewpoints of research, industry, and policymaking.

From a research perspective, We provide insights into main technological trends within the CE domain, offering a technological map, facilitating a comprehensive understanding of the

evolving technological landscape. This information is pivotal for researchers as it aids in the identification of prospective areas for research investment, and potentially yielding future synergies. Moreover, it affords researchers a clear comprehension of challenges and opportunities within specific technological fields, offering a strategic approach to research endeavours.

From an industrial standpoint, the paper provides guidelines to clarify the main technologies requisite for meaningful participation in the CE sector. It further underscores the acquisition of critical skill sets necessary for organizations to maintain competitiveness in a dynamic marketplace. Moreover, our technological map serves to R&D and production managers enabling them to make decisions concerning technology adoption and management. By providing access to associated research papers, it ensures real-time awareness of the latest developments. Additionally, the paper informs industrial stakeholders about potential implementation challenges, fostering a proactive approach to technology integration.

For policymakers, the paper provides essential insights into potential technology trends and the possible challenges linked to their practical implementation allows them to craft directives and policies for companies, ensuring their alignment with CE principles and the promotion of sustainable practices.

The paper is subject to several limitations, including the scope of observation, methodological challenges, data quality and bias issues, and limitations of text mining algorithms. Firstly, the scope mainly focuses on scientific articles that emphasize technological aspects. However, it is important to note that a significant source of information on technologies resides within patents, while scientific publications only give us the technical perspective of researchers. Identifying relevant patents in the CE domain can be challenging, and crafting effective search strategies for CE-related patents remains a complex task. Some attempts have been made to capture patents related to CE, for instance the "green patent database" (https://wipogreen.wipo.int/wipogreen-database/database), but it covers only a fraction of CE aspects, related to environmental sustainability. Furthermore, patents tend to underrepresent the economic and social sustainability facets of CE. For these reasons, one potential future development is the identification of relevant patents set for mapping the technological landscape of CE.

Secondly, the methodology adopted for technology extraction, explained in [26], presents certain precision and recall challenges. Precision concerns whether the terms we extracted effectively represent technologies. Recall pertains to whether we might have missed certain technologies. These challenges carry an element of uncertainty, since the method was originally designed for patents, not scientific articles. We have attempted to address precision issues by manually reviewing the results. To mitigate recall problems, we have sought to enhance our analysis through collaboration with domain experts, including two of the authors of this paper. Nevertheless, there is a requirement for increased efforts to effectively tackle these challenges.

Thirdly, data quality issues also pose a limitation, as the accuracy and comprehensiveness of datasets used for NLP can impact the results. In the current paper, we use a query-based approach, but the precision and recall of the query in identifying all and only the relevant scientific articles on CE is not fully explored. "Identifying all relevant articles" means that the query should retrieve every pertinent paper within the field, without missing any significant studies. Conversely, "identifying only relevant articles" means that the query should exclude any unrelated papers, ensuring that all retrieved articles are pertinent to the CE analysis. This can be problematic for the analysis because of the principle "garbage in, garbage out". Future efforts should focus on improving data quality through rigorous cleaning and validation procedures.

Finally, the limitations of NLP algorithms, such as misinterpretation of context and ambiguity in language, can also affect the results. This ambiguity issue is more pronounced in technical language, such as in scientific articles or patent documents and descriptions of

technology [26, 27], since whether a word or phrase pertains to technology heavily depends on the context in which it appears. Currently, our algorithm does not manage this aspect effectively. However, in the future, advanced NLP algorithms that rely on deep learning methods and are capable of considering context (such as BERT and other contextual word embeddings algorithms, as well as Large Language Models) are needed to fully analyse and interpret the data accurately.

Next steps include conducting a comprehensive analysis of patents related to CE technologies to capture innovations not extensively covered in scientific literature. Expanding the scope of data sources to include industry reports, grey literature, and real-world case studies can help mitigate biases and provide a more balanced perspective. Implementing rigorous data cleaning and validation procedures will enhance accuracy and reliability. Exploring advanced natural language processing techniques and machine learning models will improve precision and recall in technology extraction. Finally, engaging with experts from various fields will help refine methodologies and ensure a comprehensive understanding of CE technologies that can lead in translating research findings into practical recommendations for policymakers and industry practitioners.

## Supporting information

**S1 File. Scientific articles on circular economy.** The dataset contains title, abstract and other information of the 48,855 scientific articles retrieved using the methodology explained in Section 3.1.
(CSV)

**S2 File. Technologies with RDGR.** The dataset contains list of all technologies extracted using NLP with the growth index calculated using the RDGR, as explained in Section 3.4.
(XLSX)

**S3 File. Technological Areas in CE.** The dataset contains the technological areas in CE identified using the network analysis as explained in Section 3.4.
(XLSX)

## Author Contributions

**Conceptualization:** Vito Giordano, Alessio Castagnoli.

**Data curation:** Vito Giordano.

**Formal analysis:** Vito Giordano, Alessio Castagnoli.

**Investigation:** Vito Giordano, Alessio Castagnoli, Isabella Pecorini, Filippo Chiarello.

**Methodology:** Vito Giordano, Alessio Castagnoli.

**Project administration:** Vito Giordano.

**Supervision:** Isabella Pecorini, Filippo Chiarello.

**Validation:** Isabella Pecorini, Filippo Chiarello.

**Visualization:** Vito Giordano, Alessio Castagnoli.

**Writing – original draft:** Vito Giordano, Alessio Castagnoli, Isabella Pecorini, Filippo Chiarello.

**Writing – review & editing:** Vito Giordano, Alessio Castagnoli, Isabella Pecorini, Filippo Chiarello.

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
