## [Decision Letter · Decision Letter 0]

29 Jul 2024

PONE-D-24-14870Identifying Technologies in Circular Economy Paradigm through Text Mining on Scientific LiteraturePLOS ONE

Dear Dr. GIORDANO,

Thank you for submitting your manuscript to PLOS ONE. After careful consideration, we feel that it has merit but does not fully meet PLOS ONE’s publication criteria as it currently stands. Therefore, we invite you to submit a revised version of the manuscript that addresses the points raised during the review process.

In addition to the suggestions made by the reviewers, please note that one or more of the reviewers has recommended that you cite specific previously published works. Members of the editorial team have determined that the works referenced are not directly related to the submitted manuscript. As such, please note that it is not necessary or expected to cite the works requested by the reviewer. 

We look forward to receiving your revised manuscript.

Kind regards,

Avanti Dey, PhD

Staff Editor

PLOS ONE

Journal Requirements:

Additional Editor Comments (if provided):

Reviewers' comments:

Reviewer's Responses to Questions

**Comments to the Author**

1. Is the manuscript technically sound, and do the data support the conclusions?

Reviewer #1: Yes

Reviewer #2: Yes

2. Has the statistical analysis been performed appropriately and rigorously? 

Reviewer #1: Yes

Reviewer #2: Yes

3. Have the authors made all data underlying the findings in their manuscript fully available?

Reviewer #1: Yes

Reviewer #2: Yes

4. Is the manuscript presented in an intelligible fashion and written in standard English?

Reviewer #1: Yes

Reviewer #2: Yes

5. Review Comments to the Author

Reviewer #1: 1. Related work part has been appended at the end of the introduction section, but it is not providing a detailed review of state of the art. The authors should make a separate section/subsection on related works and present similar works. A table should be also provided to summarize the outcomes of these studies.

2. The key part of the introduction section is missing; the author should include the Contribution section. In the last paragraph, the authors must mention the organization of the paper.

3. The novelty of this paper is not clear. The difference between present work and previous Works should be highlighted.

4. The figures in the article have not been appropriately described and discussed. A brief discussion of each figure should be provided for better visibility and clarity of findings.

5. The conclusion section needs to elaborate more by discussing the disadvantages of the developed model & discussion on the results obtained. The author should also include the future work section.

6. To further enhance the literature please try to add below mentioned references

10.23919/INDIACom61295.2024.10499070

10.23919/INDIACom61295.2024.10498149

Reviewer #2: 1.Highlighting the potential contribution of the study is essential. How will the identification of these technologies contribute to advancing the circular economy agenda? Will it help in identifying gaps, fostering innovation, or guiding policy decisions?

2. Discussing the practical implications of the findings is crucial. How can the identified technologies be practically implemented in industries or integrated into existing systems to promote the circular economy?

3. Acknowledging the challenges and limitations of text mining and the study itself is important. For instance, biases in the literature, data quality issues, or limitations of text mining algorithms should be addressed.

4. To further enhance litrature add this refernce

1. https://doi.org/10.1063/5.0150432

2. https://doi.org/10.3390/su15086337

6. PLOS authors have the option to publish the peer review history of their article (what does this mean?). If published, this will include your full peer review and any attached files.

Reviewer #1: No

Reviewer #2: No

---

## [Author Response · Author response to Decision Letter 0]

14 Aug 2024

Thank you for your valuable feedback and the effort you and the reviewers have put into evaluating our manuscript. We have carefully considered all the comments and made the necessary revisions. In particular, we have included a detailed document titled "Response to Reviewers" where we have provided precise answers to each comment raised by the reviewers.

To address the reviewers' comments, we have separated the text of the review into individual comments, each highlighted in “black”. For each comment, we provide a response highlighted in “blue”, using bold text for clarity, and where necessary, we include parts of the paper's text in “blue” italic. Additionally, in the manuscript, we have tracked each change in “red” color to clearly indicate the modifications made in response to the reviewers' feedback.

We appreciate the time and effort invested by the reviewers and look forward to your further guidance.

---

## [Decision Letter · Decision Letter 1]

11 Oct 2024

Identifying Technologies in Circular Economy Paradigm through Natural Language Processing on Scientific Literature

PONE-D-24-14870R1

Dear Dr. GIORDANO,

We’re pleased to inform you that your manuscript has been judged scientifically suitable for publication and will be formally accepted for publication once it meets all outstanding technical requirements.

Kind regards,

Nazim Taskin

Academic Editor

PLOS ONE

Additional Editor Comments (optional):

Dear Authors,

I have gone through your manuscript along with the comments from the reviewers. As the comments indicate and I believe, the manuscript is well written and provides contribution to the field. While I recommend acceptance of your manuscript, I strongly suggest adding two minor points raised by the reviewers. As they are explained by the reviewers, the changes about the justification for İ) selection of period for the articles, and ii) selection of the database for searching articles from. In addition, as the reviewer suggested, a brief comparison about the results would be helpful.

Reviewers' comments:

Reviewer's Responses to Questions

**Comments to the Author**

1. If the authors have adequately addressed your comments raised in a previous round of review and you feel that this manuscript is now acceptable for publication, you may indicate that here to bypass the “Comments to the Author” section, enter your conflict of interest statement in the “Confidential to Editor” section, and submit your "Accept" recommendation.

Reviewer #3: (No Response)

Reviewer #4: All comments have been addressed

2. Is the manuscript technically sound, and do the data support the conclusions?

Reviewer #3: Yes

Reviewer #4: Yes

3. Has the statistical analysis been performed appropriately and rigorously? 

Reviewer #3: N/A

Reviewer #4: Yes

4. Have the authors made all data underlying the findings in their manuscript fully available?

Reviewer #3: Yes

Reviewer #4: Yes

5. Is the manuscript presented in an intelligible fashion and written in standard English?

Reviewer #3: Yes

Reviewer #4: Yes

6. Review Comments to the Author

Reviewer #3: After the revisions to handle first reviews, the authors are clear in terms of communicating the main purpose and contribution(s) of the study.

The paper demonstrates a good understanding of the relevant literature in the field. Quite good and recent references are cited. It adequately cites a range of literature sources, incorporating insights from various studies.

The methodology part of the study seem sufficient. However, the justification for selecting Scopus database and for selecting only one database could be provided. Similarly, the reason of excluding the papers before 2015 could also be explained.

The details and presentation of the results are good enough to provide the details and it is in line with other sections. Just one recommendation, the findings/results of the research could be compared against the ones for which limitations are listed to highlight the novalty of the existing paper in the introduction section. Despite the basic differences (e.g. comprehensivity, number of focused technologies / industries or methodology) with those papers, similarities or differences in findings could help discovering interesting inferences.

Reviewer #4: 1. Technically Sound Research:

The research described in the manuscript is technically sound. The authors used a robust methodology, including Natural Language Processing (NLP) techniques, to analyze over 45,000 scientific papers in the Circular Economy (CE) field. The data is thoroughly analyzed using well-established methods such as Named Entity Recognition (NER) and network analysis. The conclusions drawn from the data are logical and based on the results obtained from these methods. The study provides new insights into emerging technologies in CE and how they are interconnected within technological domains.

2. Data Availability:

The authors have made all data underlying the findings fully available. According to their data availability statement, the data is either contained within the manuscript and supporting information or available without restrictions. This openness ensures transparency and allows for reproducibility of the study.

3. Statistical Analysis:

The statistical analysis has been conducted rigorously and appropriately. The use of the Relative Development of Growth Rates (RDGR) metric to track the growth of technologies, along with network analysis to group related technologies, is well-founded and aligns with the research objectives. The authors applied controls, such as manual validation by experts, to ensure the accuracy of the automated extraction and classification methods.

4. Intelligibility and Writing Quality:

The manuscript is presented in a clear and intelligible manner. It is written in standard academic English, and the structure is logical, making it easy to follow the research process. The technical terminology is well-explained, and there are no significant grammatical or language issues.

5. Ethical Considerations:

There are no apparent concerns regarding research ethics or publication ethics. The manuscript does not raise issues of dual publication or conflict of interest, and the authors have declared that no competing interests exist.

Additional Comments:

• Novelty and Contribution: The manuscript makes a valuable contribution to the field by applying NLP techniques to the analysis of technological trends in the Circular Economy, a method that is less biased compared to traditional literature reviews.

• Practical Implications: The findings have practical relevance, particularly for policymakers and businesses interested in adopting emerging CE technologies.

• Limitations: The authors acknowledge the limitations of their approach, such as the reliance on academic literature and the need to expand future research to include other sources like patents.

In conclusion, the manuscript meets the standards for publication in terms of technical soundness, statistical rigor, data availability, writing quality, and ethical considerations.

7. PLOS authors have the option to publish the peer review history of their article (what does this mean?). If published, this will include your full peer review and any attached files.

Reviewer #3: No

Reviewer #4: No

---

## [Editor Report · Acceptance letter]

23 Oct 2024

PONE-D-24-14870R1 

PLOS ONE

Dear Dr. GIORDANO, 

I'm pleased to inform you that your manuscript has been deemed suitable for publication in PLOS ONE. Congratulations! Your manuscript is now being handed over to our production team.

Kind regards, 

on behalf of

Dr. Nazim Taskin 

Academic Editor

PLOS ONE